# Possible Participation of Adenine Nucleotide Translocase ANT1 in the Cytotoxic Action of Progestins, Glucocorticoids, and Diclofenac on Tumor Cells

**DOI:** 10.3390/pharmaceutics15122787

**Published:** 2023-12-16

**Authors:** Darya Ulchenko, Lilia Miloykovich, Olga Zemlyanaya, Nikolay Shimanovsky, Tatiana Fedotcheva

**Affiliations:** Science Research Laboratory of Molecular Pharmacology, Medical Biological Faculty, Pirogov Russian National Research Medical University, Ministry of Health of the Russian Federation, Ostrovityanova St. 1, 117997 Moscow, Russia; motci@list.ru (D.U.); lmiloykovich@mail.ru (L.M.); o.zemlyanaya@mail.ru (O.Z.); shimannn@yandex.ru (N.S.)

**Keywords:** gestobutanoyl, megestrol acetate, amol, dienogest, medroxyprogesterone acetate, hydrocortisone, dexamethasone, diclofenac, adenylate translocase ANT1, progestins, NSAIDs

## Abstract

A comparative analysis of the cytostatic effects of progestins (gestobutanoyl, megestrol acetate, amol, dienogest, and medroxyprogesterone acetate), glucocorticoids (hydrocortisone, dexamethasone), and diclofenac on tumor cells was carried out in order to confirm their in silico predicted probabilities experimentally. The results showed the different sensitivity of HeLa, MCF-7, Hep-2, K-562, and Wi-38 cell lines to progestins, glucocorticoids, and diclofenac. The minimum IC_50_ was found for progestin gestobutanoyl (GB) as 18 µM for HeLa cells, and varied from 31 to 38 µM for MCF-7, Hep-2, and K-562. Glucocorticoids and diclofenac were much less cytotoxic in the HeLa, MCF-7, and Hep-2 cell lines than progestins, with IC_50_ values in the range of 150–3000 μM. Myelogenous leukemia K-562 cells were the least sensitive to the action of progestins and glucocorticoids but the most sensitive to diclofenac, which showed a pronounced cytotoxic effect with an IC_50_ of 31 μM. As we have shown earlier, progestins can uniquely modulate MPTP opening via the binding of adenine nucleotide translocase. On this basis, we evaluated the expression of adenylate nucleotide translocase ANT1 (*SLC25 A4*) as a possible participant in cytotoxic action in these cell lines after 48 h incubation with drugs. The results showed that progestins differently regulated ANT1 expression in different cell lines. Gestobutanoyl had the opposite effect on ANT1 expression in the HeLa, K562, and Wi-38 cells compared with the other progestins. It increased the ANT1 expression more than twofold in the HeLa and K562 cells but had no influence on the Wi-38 cells. Glucocorticoids and diclofenac increased ANT1 expression in the Wi-38 cells and decreased it in the K562, MCF-7, and Hep-2 cells. The modulation of ANT1 expression discovered in our study can be a new explanation of the cytotoxic and cytoprotective effects of hormones, which can vary depending on the cell type. ANT isoforms in normal and cancerous cells could be a new target for steroid hormone and anti-inflammatory drug action.

## 1. Introduction

Progestin preparations, in particular, the pregnane progestins medroxprogesterone acetate (MPA) and megestrol acetate (MA), have antitumor activity and are indicated for the treatment of cervical and endometrial cancers [1,2,3]. These progestins have anticachexic [4,5], anti-inflammatory [6,7], and direct antiproliferative action on hormone-dependent tumors [8,9,10]. The novel pregnane progestins gestobutanoyl and its metabolite amol, as well as the anti-inflammatory chimeric progestin dienogest, are potential anticancer drugs: antitumor activity has already been shown for gestobutanoyl in vivo [11], and the cytotoxic activity of dienogest on human endometrial cancer cells and breast cancer cells MCF-7 was demonstrated previously [12,13], but progestin dienogest has no indications for cancer treatment.

The ability of steroid hormones of different classes to inhibit the proliferation of cancerous cells is poorly studied. Meanwhile, there is a class of steroid hormones with anti-inflammatory action, which has implications for cancer treatment—glucocorticoids (GCs). They are commonly used in anticancer therapy, especially in the treatment of lymphomas and leukemias [14,15]. The mechanisms of their anticancer action include the transactivation of apoptosis-inducing genes, such as *Bim* (Bcl-2 Interacting Mediator of cell death) [16], and the inhibition of *AP-1-*, *c-MYC-*, and *NF-****κ****B*-mediated transcriptions [17]. Dexamethasone treatment induces the transcription of factor Runx2 and c-Jun in parallel with *BIM* induction [18]. The anticancer effectiveness of GCs is well established for hematological malignancies [19], but there are controversial data on their influence on solid tumors. GCs may cause the progression of highly aggressive pancreatic ductal adenocarcinoma (PDAC) by inducing miR-378 after binding with glucocorticoid receptors [20].

Since the pathways of inflammation signaling and carcinogenesis usually intersect [21], NSAIDs (non-steroidal anti-inflammatory drugs), whose typical representative is diclofenac, have potential in the treatment of tumors. Recently, the cytotoxic action of diclofenac has been established on different cancerous cell lines during 48 h incubation with the following IC_50_ values: MCF-7, 150 µM; HeLa, 548 µM; and HT-29, 248 µM [22]. Diclofenac has a cytotoxic effect and enhances the cytotoxic action of cisplatin on A549 human lung adenocarcinoma cells and SBC-3 human small cell lung cancer cells [23]. Recently, the role of COX-2 and NF-kB expression has been established in cancer drug resistance [24]. Because the mechanism of the anti-inflammatory action of diclofenac and GCs is due to COX-2 and NF-kB inhibition, this could be one of the reasons for their cytotoxic effect on cancerous cells. Diclofenac has also antiangiogenic activity [25]. Similarly, glucocorticoids have both anti-inflammatory and antitumor activities [17]. Dexamethasone (DEX) demonstrated an anticancer effect by damaging the DNA and inducing oxidative stress in the A549 cancer cell line [26]. High-dose DEX inhibits the invasion of blood vessels, the levels of the cell proliferation markers Ki67 and c-Myc, and the antiapoptotic marker Bcl-2. The activation of M1-like tumor-associated macrophages and inefficient glucose and lipid metabolism are also mechanisms of delayed tumor cell growth and apoptosis promotion during DEX treatment [27]. Both GCs and NSAIDs downregulate inflammation via COX and NF-kB inhibition as well as proinflammatory cytokines, which can lead to STAT3 inhibition and, accordingly, the inhibition of cell proliferation and invasion [28]. The anti-inflammatory action of different drugs underlies their cytotoxic effect: both glucocorticoids and nonsteroid anti-inflammatory drugs (NSAIDs) may be effective in cancer treatment. The cytotoxic and anti-inflammatory activities of the pregnane progestins MPA, MA, GB, and amol have been predicted using in silico analysis, which showed their higher cytotoxic action compared with dienogest. One of the goals of this study was to experimentally confirm this prediction.

Until now, the cytotoxic activity of progestins, glucocorticoids, and diclofenac has not yet been evaluated on Hep-2 and K-562 cancer cell lines. There is no information about the IC_50_ values for progestins and glucocorticoids, which could add to knowledge of the pharmacological activity of these hormones and expand their indications for use, since the repurposing of drugs, in particular diclofenac [29], has become a frequent occurrence.

In this work, in order to identify the progestin with the highest cytotoxic effect, their influence on the viability of HeLa, MCF-7, Hep-2, and K-562 tumor cells has been studied. In order to assess the selectivity of the cytotoxicity of progestins, their influence on the viability of normal cells (human lung fibroblasts WI-38) was also evaluated.

The importance of studying new pharmacological properties of progestins is due to the possibility of their usage as monotherapy drugs in the treatment of hormone-dependent tumors [3,8,30], as well as due to their anti-inflammatory and immunomodulatory action in the treatment of cancer-associated inflammation [7,31,32].

The mechanism of the cytotoxic action of progestins can be realized not only via known signaling [33], but also via the regulation of the MPTP component, adenine nucleotide translocase (ANT), as it has been previously shown that progestins unequally regulate its function [34], and molecular docking demonstrated a high binding energy comparable with that of carboxyatractilozide, a specific regulator of ANT [35]. The difference in the action of steroids on cancerous and noncancerous cells may be due to the energy uptake processes in cells. The energy in a healthy cell is obtained from oxidative phosphorylation in the mitochondria, and cancer cells can use aerobic glycolysis for extensive proliferation due to hypoxic adaptation [36]. A direct role in these processes belongs to ANTs.

Previously, we have revealed the specific effect of pregnane progestins on the induction of MPTP opening via calcium ions, depending on the chemical structure of the progestin. The C3 derivatives of progesterone (one of them is GB in this study) had a pronounced concentration-dependent inhibitory effect on pore opening, whereas progesterone and MPA stimulated MPTP opening [34].

The aim of this study was to evaluate the cytotoxic action of the C3 derivative of the progestin GB and other steroid hormones and compare this action with the expression of ANT1 as a new possible mechanism of their cytotoxicity.

## 2. Materials and Methods

### 2.1. Materials

The following active substances were used in the study: the progestin gestobutanoyl (GB) and its metabolite amol were synthesized at the Pirogov Russian National Research Medical University. The structural formulas of GB and amol are shown in Figure 1. The other compounds tested on the cells, medroxyprogesterone acetate (MPA), megestrol acetate (MA), and dienogest (D), as well as the reference substances glucocorticoid hormones dexamethasone (DEX) and hydrocortisone (HC), and the non-steroidal anti-inflammatory drug diclofenac (DFC), were obtained from Sigma (St. Louis, MO, USA).

The following reagents and equipment were used: culture flasks (Eppendorf, Hamburg, Germany); 96-well plates (Corning and Costar, Cambridge, MA, USA); DMEM/F-12 cell culture media with glutamine and RPMI (Gibco, London, UK); fetal bovine serum 10% (Gibco, Auckland, New Zealand); penicillin-streptomycin, trypsin, Versen solution, saline solution, dimethyl sulfoxide (DMSO) (PanEco, Moscow, Russia); amphotericin B (Synthesis AKOMP, Moscow, Russia); 3-(4,5-dimethylthiazol-2-yl)-2,5-tetrazolium bromide (MTT) (Diam, Moscow, Russia); a Uniplan AIFR-01 plate photometer (Ryazan, Russia).

### 2.2. In Silico Analysis

The PASS Online program (http://www.way2drug.com/ (accessed on 20 October 2023) was used to predict the biological activities of the substances, based on their SMILES structures, obtained from the PubChem database.

### 2.3. Cell Viability Assay

In this study, the following cell lines were used: human cervical adenocarcinoma HeLa, human chronic erythroleukemia K-562, human larynx carcinoma Hep-2, human breast adenocarcinoma MCF-7, and a culture of non-tumor cells—human embryonic lung fibroblasts Wi-38. All cell lines were obtained from the Russian Research Institute of Medicinal and Aromatic Plants biocollection bank: http://vilarnii.ru/biokollektsii/ (accessed on 30 August 2023).

The cell cultivation was carried out under sterile conditions on the culture medium DMEM/F-12 (for HeLa, Hep-2, MCF-7, Wi-38) and RPMI (for K-562) supplemented with 10% fetal calf serum and antibiotics (penicillin and streptomycin 100 units/mL, amphotericin B 100 µg/mL). The cell cultures were grown at 37 °C in an atmosphere of 5% CO_2_ and 100% humidity.

The cell viability was assessed using the MTT method [37]. The cell cultures were seeded in 96-well plates at a density of 10,000 cells per well. After the cells were attached to the surface of the wells, they were incubated in the presence of the test substances at final concentrations in the range of 10^−8^ M to 10^−4^ M for 48 h. The DMSO concentration did not exceed 0.01%, and the control wells contained an equal volume of DMSO at each point. At the end of the incubation, the medium in which the cells with substances were cultivated was replaced with a nutrient medium without fetal calf serum, which contained 5 mg/mL MTT, and the incubation continued for another 2 h in a CO_2_ incubator. After incubation, the medium with MTT was removed, and a DMSO solution was added to the wells to dissolve the formazan precipitate formed from MTT during the cell life. The optical density of the samples was measured at a wavelength of 530 nm on a photometer Uniplan AIFR-01 (Pikon, Moscow, Russia). The cell viability in the cultures was expressed as a percentage relative to the values obtained for the control samples (without tested compounds). The average values of optical density in the control samples were taken as 100%.

### 2.4. Real-Time PCR

The cell pellets (1 × 10^6^ cells) of each cell line were collected after incubation in 125 mL flasks with the appropriate steroid or diclofenac at a fixed concentration of 10 μM. The mRNA was extracted using an RNA extraction kit (Syntol, Moscow, Russia), and cDNA synthesis was performed using the cDNA Reverse Transcription Kit OT-1 (Syntol, Russia) with random hexamers according to the manufacturer’s instructions. The real-time steps were run on an iCycler iQ5 real-time PCR instrument (Bio-Rad, Hercules, CA, USA) in 40 cycles, with the following steps: denaturation step (10 s, 95 °C), annealing and elongation step (40 s, 62 °C). We used 10 μL of the quantitative real-time PCR reaction mix from a commercial kit (Cat. No M-427, Syntol, Moscow, Russia) for each probe. Primers were used at 100 nM for each probe per reaction, 2 μL of the forward primer, and 2 μL of the reverse primer. The total probe volume in a flat cap was 25 μL. The Gapdh (glyceraldehyde phosphate dehydrogenase) gene was used as a control gene (housekeeping gene). The primer sequences were as follows: *GAPDH*: *gaa-ggt-gaa-ggt-cgg-agt* (up), *gaa-gat-ggt-gat-ggg-att-tcc* (low); *SLC25A4*, adp-atp translocase 1: *agt-ggc-tca-tgc-ctg-taa-tc* (up), *gag-agg-agg-tct-tgc-tat-gttg* (low). To determine the levels of gene expression, the values 2^−∆∆CT^ were used, where the threshold number of cycles (Ct) is the number of PCR cycles at which the fluorescence exceeds the threshold value (to detect significant differences between data groups) and 2^−∆∆CT^ (for determining the multiplicity of differences, where ∆Ct = Ct (of the desired gene) − Ct (GAPDH) and ∆∆Ct = ∆Ct (sample) − ∆Ct (control)) [38].

### 2.5. Statistical Analysis

Statistical analysis was carried out using the GraphPad Prism v8.4.3 software with nonlinear regression analysis for the cell viability assay. Each experiment was repeated three times, with three repetitions (3 wells) within each experiment and 12 repetitions for the control wells (for the control in each experiment, the whole line of the 96-well plate was used to avoid large differences within the control points). The mean ± standard deviation value was calculated for each point.

The statistical significance of the real-time PCR data was determined using the Mann–Whitney U test (*p* < 0.05) between the control group (non-treated cells) and treated group (drug-treated cells). Each experiment was repeated three times, with three repeats (3 wells) within each experiment. The mean ± standard deviation value was calculated for each point.

## 3. Results

### 3.1. PASS Online Prediction of the Cytotoxic Action of Progestins, Glucocorticoids, and Diclofenac

The differences in the cytotoxic action of progestins, GCs, and DCF and the possible impact of their anti-inflammatory activity were evaluated using the PASS online program [39]. The probabilities (Pa) for the anti-inflammatory and cytotoxic activities in the PASS online program for the compounds studied are presented in Table 1.

The in silico analysis demonstrated very high probabilities for the anti-inflammatory activity for all the drugs studied: progestins, GCs, and DCF. But the probability of cytotoxic activity was high for the pregnane progestins GB, MA, amol, and MPA, low for the progestin dienogest (D), and was not predicted for DEX and DFC—see Table 1.

The cytotoxic activity of the pregnane progestins was confirmed experimentally on four cancerous cell lines.

### 3.2. Cytotoxic Action of Progestins, Glucocorticoids and Diclofenac on Tumor Cells

For the estimation of the cytotoxic effect of the hormones and DCF, cells were incubated for 48 h with the tested compounds in a concentration range from 10^−8^ to 10^−4^ M. The data are shown in Figure 2.

The dependence of the cell viability on the concentration of the substances studied is presented on Figure 2.

The analysis of the experimental curves using the GraphPad Prism v8.4.3 program (San Diego, CA, USA) gave the half-maximal inhibition concentrations (IC_50_) for each compound on all cell lines. The results are shown in Table 2.

The cytotoxic action of the studied compounds was more pronounced toward tumor cell lines than toward the embryonic fibroblasts Wi-38. The progestins had lower IC_50_ values for the tumor cells than the glucocorticoid hormones and DCF on adherent cell cultures (Table 2). DCF was the most cytotoxic compound (IC_50_ = 31 μM) for the suspension culture K562. The pregnane progestins MPA, GB, and MA were more cytotoxic than D.

According to the IC_50_ data for each compound, the selectivity index (SI) of the cytotoxic action was determined using the following formula: IC_50_ (in non-tumor Wi-38 cells)/IC_50_ (in tumor cells) [40]. The SI values are presented in Table 3.

Among the hormones, only DEX, with an SI = 16.71, produced selective suppression of K-562 cell proliferation. At the same time, the NSAID diclofenac had a pronounced cytotoxic activity only on these tumor cells; this means that the use of anti-inflammatory drugs is an important element of pathogenesis treatment of chronic myelogenous leukemia. Considering the data on a maximum concentration of DCF in the blood of 5 μM by 2 h after the oral administration of a 50 mg tablet, the experimentally obtained IC_50_ of 31.5 μM is in a range that is easily achievable for humans. Such a concentration should not cause dramatic side effects. DCF has recently been shown to have a cytotoxic effect by increasing ROS production and causing microtubule destabilization in HeLa tumor cells [30]. The IC_50_ for DCF on the HeLa cells obtained in our study is 384 µM, which is in the same range as the values obtained earlier, 200 and 548 µM [22,41].) The IC_50_ obtained on the HeLa cells is significantly higher than that obtained for K562 cells, 31.5 µM.

The cytotoxic action of the drugs studied can be due to their anti-inflammatory action (Table 1) since there is a positive correlation between their predicted anti-inflammatory and cytotoxic properties (r = 0.57). All the cell lines tested are hormone-sensitive. The difference in the cytotoxic action of the hormones and DCF involves an unknown mechanism, which could be related to the different energy metabolism of the cells and particularly to adenine nucleotide transport in the mitochondria.

### 3.3. The Influence of Progestins, GCs, and DCF on the Expression of Adenine Nucleotide Translocase ANT1 in Tumor Cells

In order to confirm the previously obtained data on the specific regulation of MPTP opening by pregnane progestins and the predicted data on the high energy of binding with one of the components of MPTP, ANT1, here we compared the influence of the progestins and DCF on ANT1 expression in the same cell lines during 48 h incubation at a fixed concentration of 10 μM. The data are shown in Figure 3.

In the MCF-7 cells, all the tested compounds downregulated the ANT1 expression. GB was the only one compound with an ester at C3 that uniquely affected the ANT1 mRNA expression: it decreased it in MCF-7 more than fourfold, increased it more than twofold in the HeLa and K562 cells, and had no influence on the Wi-38 cells. In the MTT test, GB was also the most active cytotoxic agent. The MCF-7 cell line is sensitive to progestins: P_4_ was shown to inhibit the proliferation of these cells with an IC_50_ value of 6.5 ± 0.2 μM [42]. The anticancer action of progestins is known to be realized via the membrane and nuclear progesterone receptors (PR) [43]. Since all the drugs tested downregulated the ANT1 expression in MCF-7, it can be assumed that ANT1 downregulation in PR-positive cells can lead to cell death.

In normal Wi-38 cells, all compounds except GB stimulated the ANT1 expression. As ANT1 plays a positive cytoprotective role in normal cells (for example, cardiomyocytes [44]), this result could mean that all the drugs tested are positive regulators of ATP exchange processes in lung fibroblast mitochondria and have antiapoptotic action, as ANT1 is linked with BCL-2 proteins.

The most pronounced effect of the tested drugs on ANT1 expression was demonstrated in the K562 cells. These results show a good correlation between the IC_50_ value of DCF and the ANT1 expression in these cells: the lowest value for diclofenac IC_50_ is in good agreement with the highest inhibition rate of ANT1 expression (r = 0.75).

On the Hela cells, the most pronounced effect—in the stimulation of ANT1 expression—was shown for GB and its metabolite amol. Recently, it has become obvious that the effect of potential antitumor agents, and in particular steroid hormones, depends on the cell type and the cellular metabolic phenotype, since they all affect the glucose metabolism, lactate production, lipid metabolism, and the cellular redox status [45].

## 4. Discussion

To evaluate the participation of ANT1 in the regulation of cell viability by steroid hormones and DCF, we examined their effect on the viability and ANT1 expression in cancerous cell lines sensitive to steroid hormones. As is known, all the cell lines, HeLa [46], MCF-7 [43], Hep-2 [47], and K-562 [48], contain steroid hormone receptors. It is known that PR and GR are expressed in the lung fibroblasts [49]; however, information about the expression of PR or GR in the Wi-38 cell line is not available.

The IC_50_ values obtained in this work for the progestins GB and MA on the tumor cells are much lower than the IC_50_ for GCs and DCF. The IC_50_ range for progestins is within the limits of hormone concentrations achievable in the blood during long-term treatment of endometriosis and endometrial hyperplasia [6]. The previously obtained IC_50_ values for GB, MA, P_4_, and MPA after 72 h incubation were somewhat different from those obtained during 48 h incubation on HeLa cells—72 h: GB 9.6 µM, amol 30 µM, MA 200 µM; 48 h: GB 18 µM, amol 31 µM, MA 33 µM—which is explained by the metabolism of GB in HeLa cells: actually, amol and MGA are direct metabolites of GB [50], which provide prolonged GB action compared with MA. During 72 h incubation with the cells, the cytotoxic action of MA and amol is coupled with that of GB; thus, all three compounds demonstrate an additive effect. The action of MA itself is reduced during incubation.

The detected ability of DCF to suppress the viability of the K-562 cells could be taken into account in the treatment of lymphocytic leukemia. Some well-known NSAIDs are already being repurposed: they can be used not only in the treatment of cancer-related pain but also as anticancer drugs and chemosensitizing drugs [24,41,51]. The cytotoxic action of NSAIDs is realized via different mechanisms: a decrease in the expression of MYC (a family of regulator genes and protooncogenes that code for transcription factors), the regeneration of intracellular ROS and induction of apoptotic death, cell cycle arrest at different checkpoints, the inhibition of cell proliferation and migration, activating autophagy, and microtubule destabilization [29,52]. DCF also sensitizes cancerous cells to cisplatin and 5-fluorouracil [41,52]. The chemosensitizing activity of NSAIDs is due to the inhibition of both the expression and enzyme activity of SOD2 and COX-2. SOD2 protects against ROS, while increased COX-2 levels were found to be the reason for enhanced cellular proliferation and apoptosis resistance in various cancers [53,54]. In the concentration range of 0.1–0.8 mM, which is similar to that in our study, diclofenac inhibits decreased glucose transporter 1 (GLUT1), lactate dehydrogenase A (LDHA), and monocarboxylate transporter 1 (MCT1) gene expression, and lowers the secretion of lactate in leukemia and melanoma cell lines [55]. Thus, diclofenac affects not only the lactate transporter but also the adenine nucleotide transporter, and these two participants can provoke cell death.

Here, we showed that the IC_50_ values for DCF and GCs for the cancerous cells were less than the IC_50_ for the normal Wi-38 cells. The selectivity of DCF and GC toward cancerous cells is due to their anti-inflammatory and antineoangiogenic action: decreased synthesis of prostaglandins attenuates both of these activities. An important mechanism of the selective cytotoxic action of DCF is its influence on glycolysis in tumor cells: it was shown that DCF inhibits lactate transporter activity with an IC_50_ of 1.45 ± 0.04 μM for MCT1 and 0.14 ± 0.01 μM for MCT4 [56].

The novel mechanism of the cytotoxic action of DCF may be related to the differential regulation of the expression of adenine nucleotide transporter ANT1. As follows from our data, DCF modulates the ANT1 expression in cancerous and noncancerous cells in the same manner as the steroid hormones.

As is known, the specificity of the cytotoxic action of progestins is due to the inhibition of survival-signaling pathways, especially the TGF-β and Wnt/β-catenin pathways [33]. Progestins are also able to increase the antiproliferative action of cytostatic drugs by inhibiting the P-glycoprotein expression and activity [33] and due to their own cytotoxic activity, which is realized via nuclear and membrane progesterone receptors [57].

The differential regulation of ANT expression is also a novel property of progestins. Previously, we have shown that pregnane progestins have a specific effect on the induction of MPTP opening via calcium ions, which depends on the chemical structure of progestin. The C3 derivatives of progesterone (one of them is GB in this study) had a pronounced concentration-dependent inhibitory effect on pore opening, whereas progesterone and MPA stimulated MPTP opening. The inhibitory effect of C3 derivatives was eliminated in the presence of carboxyatractyloside, a selective inhibitor of adenylate translocase [34]. Subsequently, using molecular docking, it was shown that pregnane ligands strongly bind to the target (ADP/ATP translocase). The minimum free energies of the conformations of all tested substances ranged from −7.22 to −11.38 kJ/mol. The control ANT inhibitor carboxyatractyloside binds to targets very strongly (the range −16.12 to −17.09 kJ/mol) [35].

ANTs have four isoforms (ANT1-4); the most studied and abundant of these is ANT1, which is encoded by the *SLC25 A4* gene. A comparison of the experimental data and computed results showed the trend in the dependence between the IC_50_ value and the binding energy value for ANT1 [35]. ANT1 serves as a cardiac- and skeletal-muscle-specific ATP transporter involved in mitochondrial DNA maintenance and apoptosis; in cancer cells, its overexpression can sensitize to chemotherapeutic drugs and induce apoptosis [58]. In normal cells, ANT1 is an important component in antioxidative cell-protective processes. ANT1 is part of the cardioprotective TLR4 signaling mediated by HSP27 [44].

The role of ANT1 is still unclear. There are controversial data on the level of expression and dependence of cell death under different conditions. On the one hand, ANT1 expression is lowered in patients with severe cervical carcinoma and prostate cancer: overexpression promotes apoptosis in cultured breast cancer cells [59]. On the other hand, in cancerous human glioblastoma cells, the silencing of ANT1, but not ANT2, strongly reduced viability by inducing an increase in oxidative stress, which led to cell death [60,61]. ANT1 is considered to be a proapoptotic isoform for some tumor cells [61] and antiapoptotic and cytoprotective for cardiomyocytes [44]. The overexpression of adenylate translocase and a reduction in the expression of detoxifying enzymes can enhance the sensitivity of cancer cells to apoptotic and oxidative stimuli [62,63]. This is shown in our study for GB. The role of ANT1 expression is still ambiguous; in tumor cells, ANT1 expression can lead to both proapoptotic and antiapoptotic stimuli. The regulation of ANT1 expression may be another novel mechanism for the cytotoxicity of pregnane steroids and diclofenac on tumor cells.

The comparison of progestins in terms of their inhibitory effects on MPTP opening and the differential regulation of ANT1 reveals a novel approach to decreasing the cytoxocity of the chemotherapeutic drugs toward normal cells and its increase toward cancerous cells. C3-substituted progestins can specifically bind the SH groups of ANT1 and, at the same time, the SH groups of the nucleotide-binding domain of MDR transporters [34]. Thus, these agents may be effective as MDR modulators and chemosensitizers, when used together with first-line therapy drugs such as doxorubicin.

## 5. Conclusions

The in silico predicted cytotoxic activity of progestins, glucocorticoids, and diclofenac has been experimentally confirmed on hormone-dependent Hep-2, HeLa, MCF-7, and K-562 cancer cell lines, which contain both progesterone and glucocorticoid receptors.

A significantly more pronounced cytotoxic effect on cancerous cells was observed with progestins rather than with glucocorticoids and diclofenac. The K-562 cells were less sensitive to the cytostatic action of progestins than the HeLa, Hep-2, and MCF-7 cells, but more sensitive to the action of dexamethasone and diclofenac. Diclofenac had the most pronounced cytotoxic effect on the K562 cells (31.5 µM). In the last decade, diclofenac has been intensively studied as an anticancer drug. NSAID administration in the palliative therapy of advanced cancer could be a promising strategy in the treatment of hematologic malignancies.

Here, for the first time, the specific regulation of ANT1 expression by progestins, glucocorticoids, and diclofenac has been demonstrated, which suggests the involvement of this mitochondrial transporter in the mechanism of the cytoprotective and cytotoxic action of the hormones depending on the cell type. Gestobutanoyl cytotoxic action is linked more tightly with ANT1, but diclofenac has different mechanisms, one of which is also connected with ANT1.

The modulation of ANT1 expression can be a new mechanism of the cytotoxic and cytoprotective effects of steroid hormones and anti-inflammatory drugs depending on the cell type. In cancerous cells with disrupted energy consumption, ANT could be the target of their specific action leading to cell death. Further studies are required to determine the specific role of ANT isoforms in cancer and inflammation. The identification of selective modulators of ANT expression among the old drugs can help to better repurpose them and find better combinations with new drugs. Future studies should aim at testing first-line chemotherapeutic drugs for the expression of ANT isoforms and their levels, as well as their dependence on the stage of cancer disease and the dynamics of the changes upon treatment. The participation of ANT isoforms in proliferation, differentiation, and cell metabolism requires deep investigation to determine their role in these processes. The signaling via ANT isoforms may be a new target of steroid hormones and anti-inflammatory drugs. For all drugs tested, we found a strong correlation between the cytotoxic effects of the pregnane progestins GB, MA, and amol and their effects on the ANT1 expression in MCF-7 and Wi-38 cells (r = 0.99 and 0.93). The results indicate the specific regulation of ANT1 expression and the impact of this mitochondrial transporter on their cytotoxic action. Inhibitors of ANT1 expression can significantly increase the cytotoxic effect of chemotherapeutic agents and simultaneously reduce their toxic effect on the mitochondria of heart, liver, and kidney cells. The development of therapeutic strategies targeting ANT1 is thus an important issue for future pharmacology.

## Figures and Tables

**Figure 1 pharmaceutics-15-02787-f001:**
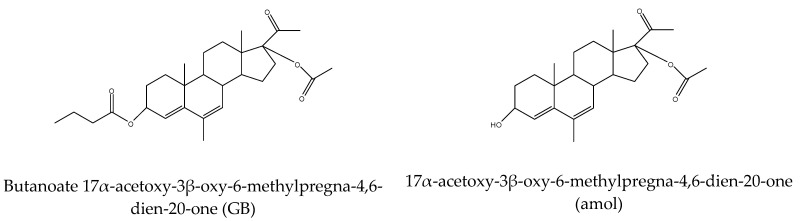
Structural formulas of the progestins GB and amol.

**Figure 2 pharmaceutics-15-02787-f002:**
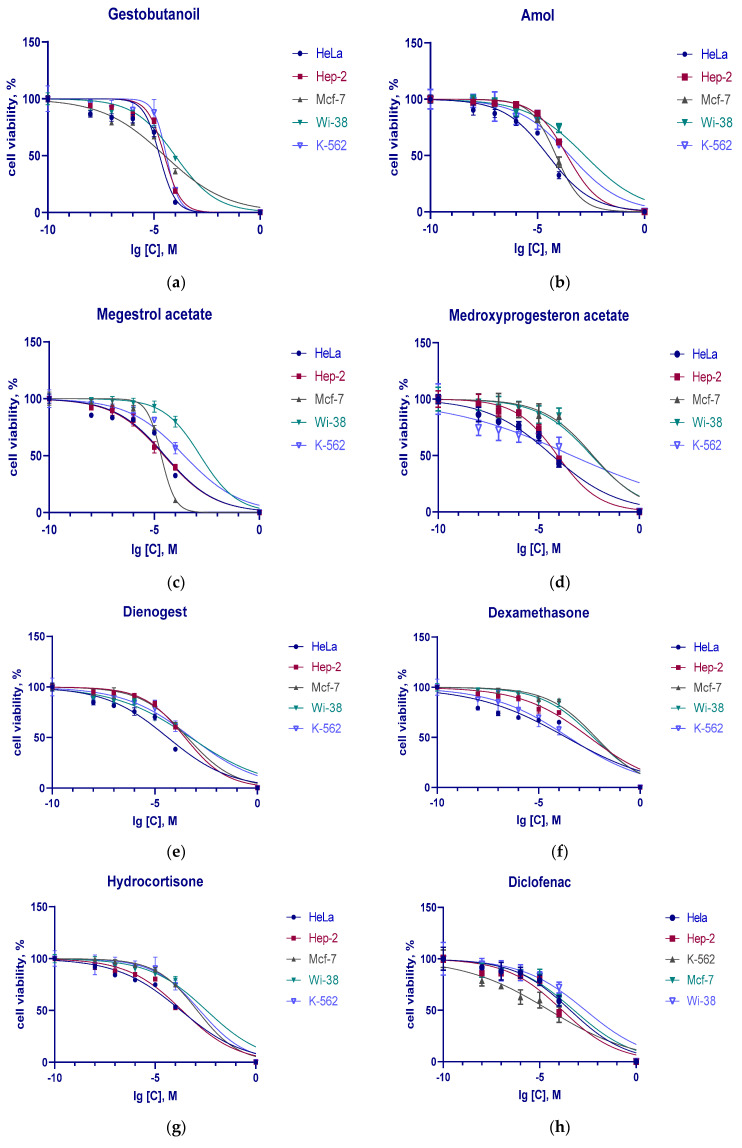
The effect of gestobutanoyl (**a**), megestrol acetate (**b**), amol (**c**), medroxyprogesterone acetate (**d**), dienogest (**e**), dexamethasone (**f**), hydrocortisone (**g**), and diclofenac (**h**) on the viability of cancerous (HeLa, MCF-7, Hep-2, K-562) and noncancerous (Wi-38) cells after 48 h incubation. Note: the values are presented as the means (M) normalized to control ± standard error of the mean (s.e.m.). The control wells contained an equal volume of DMSO at each point.

**Figure 3 pharmaceutics-15-02787-f003:**
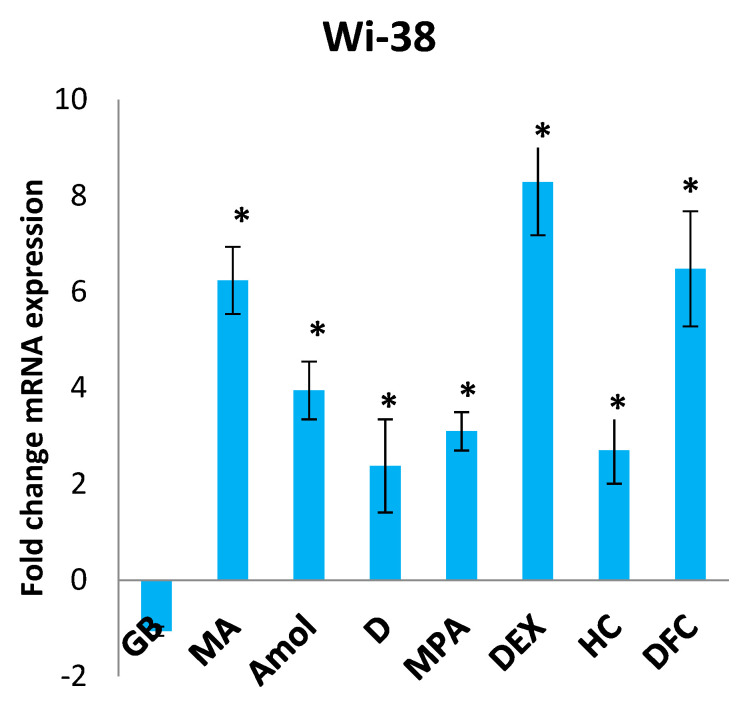
The influence of progestins, GB, and DFC at a fixed concentration of 10 μM on the expression of adenine nucleotide translocase ANT1 in cancerous (HeLa, MCF-7, Hep-2, K-562) and noncancerous (Wi-38) cells after 48 h incubation. Note. *—significant (from 2- to 12-fold) change in gene expression. *p* < 0.05.

**Table 1 pharmaceutics-15-02787-t001:** Prediction of the cytotoxic and anti-inflammatory activity for progestins, glucocorticoids, and diclofenac.

Predicted Activity	GB	MA	Amol	D	MPA	DEX	HC	DFC
Anti-inflammatory	0.785	0.84	0.811	0.597	0.883	0.985	0.94	0.791
Cytotoxic (antitumor)	0.814	0.767	0.774	0.532	0.727	-	0.679	-

Note: Pa (probability of activity) values are given in the table. If Pa > 0.7, the substance is very likely to exhibit the activity, but the chance of the substance being the analog of a known pharmaceutical agent is also high [39].

**Table 2 pharmaceutics-15-02787-t002:** IC_50_ values (μM) for the tested substances after incubation with cancerous (HeLa, MCF-7, Hep-2, K-562) and noncancerous (Wi-38) cells for 48 h.

	GB	MA	Amol	D	MPA	DEX	HC	DCF
HeLa	18.3 ± 2.18	32.9 ± 3.29	31.3 ± 4.7	44 ± 4.8	49.9 ± 8.73	378 ± 109.24	215 ± 32.3	384 ± 99.1
Hep-2	31 ± 4.96	29.2 ± 5.0	238 ± 39.7	296 ± 44	85.1 ± 14.21	3160 ± 543	217 ± 30.16	146 ± 22.5
MCF-7	31.9 ± 5.17	21.3 ± 2.85	73.7 ± 12.7	387 ± 61.53	4910 ± 85.3	6540 ± 1294.9	1230 ± 212.79	595 ± 119.0
K-562	38.2 ± 8.4	266 ± 62	326 ± 60	611 ± 123	441 ± 121.72	304 ± 63.8	1660 ± 417	31.5 ± 8.45
Wi-38	98.9 ± 21.1	1540 ± 297	1620 ± 460.1	546 ± 115	4200 ± 721.4	5080 ± 1132.8	3530 ± 687.2	2330 ± 459.4

Note: GB, gestobutanoyl; MA, megestrol acetate; D, dienogest; MPA, medroxyprogesterone acetate; DEX, dexamethasone; HC, hydrocortisone; DCF, diclofenac.

**Table 3 pharmaceutics-15-02787-t003:** Selectivity index (SI) for progestins, glucocorticoids, and diclofenac toward Wi-38 cells.

Drug Name	SI			
	HeLa	Hep-2	MCF-7	K-562
GB	5.40	3.19	3.10	2.59
MA	46.81	52.74	72.30	5.79
A	51.76	6.81	21.98	4.97
D	12.41	1.84	1.41	0.89
MPA	84.17	49.35	0.86	9.52
DEX	13.44	1.61	0.78	16.71
HC	16.42	16.27	2.87	2.13
DCF	6.06	15.96	3.92	74

Note: GB, gestobutanoyl; MA, megestrol acetate; A, amol: D, dienogest; MPA, medroxyprogesterone acetate; DEX, dexamethasone; HC, hydrocortisone; DCF, diclofenac. SI values less than 1 indicate a lack of selectivity toward tumor cells; a larger SI indicates a more pronounced effect of the compound on tumor cells.

## Data Availability

The data presented in this study are available in this article.

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
