# Peer review of "Possible Participation of Adenine Nucleotide Translocase ANT1 in the Cytotoxic Action of Progestins, Glucocorticoids, and Diclofenac on Tumor Cells"

_pharmaceutics, 2023, doi:10.3390/pharmaceutics15122787_

Round 1

Reviewer 1 Report

Comments and Suggestions for Authors

Comments to the Authors 

This article titled Possible participation of adenine nucleotide translocase ANT1 in the cytotoxic action of progestins, glucocorticoids, and diclofenac on tumor cells. The authors mainly compared the cytostatic effects of progestins (gestobutanoyl, megestrol acetate, amol, dienogest, and medroxyprogesterone acetate), glucocorticoids (hydrocortisone, dexamethasone), and diclofenac on tumor cells.

The language and arrangement of article is not good. This article can be accepted after minor revision.However, there are still have some questions.

1. The quality of Figures and Tables shoud be improved.

2. The authors should add the further opinion in the further research.

3. The whole Abstract and Discussions has many contents, authour should shorten them.

Thus, this article can be accepted after minor revision.

Reviewer 2 Report

Comments and Suggestions for Authors

This study investigated the cytotoxic action of progestins, glucocorticoids, and diclofenac on tumor cells and their regulation on adenylate nucleotide translocase ANT1 (SLC25 A4) using the human embryonic lung fibroblasts Wi-38 cell line and various other cancer cell lines. The authors’ results suggest that these cytotoxic effects may be related to their regulation of the ANT1 gene expression. This reviewer identified the following issues with this manuscript.

1)      The authors need to rewrite this manuscript in a more concise way for all sections. For example, the abstract (379 words) is too long with three paragraphs.

2)      The legends of all figures are too short to stand alone. More information is necessary to expand these legends.

3)      The impact of all drugs on ANT1 gene was assessed only at the mRNA level; it is necessary to assess whether they also affect ANT1 protein expression such as using western blotting assays

4)      As no statistical analysis information provided in the figure 3 legend, not  sure which statistical method they used to examine the statistical significance of their findings.

Comments on the Quality of English Language

rewriting in a more concise  way is necessary

Reviewer 3 Report

Comments and Suggestions for Authors

1.            Mention the novelty or significance of the study findings in the abstract.

2.            The transition between discussing progestins and glucocorticoids to diclofenac is abrupt. Consider adding a sentence to smoothly introduce the role of NSAIDs like diclofenac in cancer treatment.

3.            The rationale for evaluating the cytotoxic activity on specific cancer cell lines (Hep-2 and K-562) is well stated. However, it might be helpful to briefly mention why these cell lines were selected and what unique insights their study might provide.

4.            Consider providing additional details on the Real-Time PCR method, such as the cycling conditions, reaction volumes, and the specific commercial kit used for the quantitative real-time PCR reaction mix.

5.            In the Real-Time PCR subsection, consider providing information on the selection and validation of the reference gene (GAPDH) and discussing its stability in the context of the study.

6.            In the statistical analysis subsection, specify the parameters measured by non-linear regression analysis and the rationale behind choosing this analysis method.

7.            Clarify whether the Mann-Whitney U-test was used for comparing groups and provide information on the groups compared (e.g., control vs. treatment).

8.            The explanation of 12 repetitions for the control wells is not entirely clear. Consider revising this sentence for clarity.

9.            Consider providing statistical measures or variability (e.g., standard deviation) for the IC50 values in Table 2 to indicate the reliability of the measurements.

10.         When discussing the effects on ANT1 expression (Figure 3), consider integrating this information with the cytotoxicity data. For instance, discuss whether changes in ANT1 expression correlate with the observed cytotoxicity.

11.         In the last paragraph, where the effect of potential antitumor agents is discussed, consider providing more context or specific examples to enhance the reader's understanding.

12.         In the discussion of DCF's cytotoxic action, consider providing more context on the relevance of inhibiting lactate transporters and lowering lactate secretion by tumor cells. Explain how these actions might contribute to the observed cytotoxicity.

13.         The discussion on the regulation of ANT1 expression is intriguing. Consider expanding on the potential implications of differential ANT1 regulation in cancerous and noncancerous cells. For example, discuss how these findings might inform future research or therapeutic strategies.

14.         The comparison of progestins in terms of their inhibitory effects on MPTP opening and the differential regulation of ANT1 is a valuable addition to the discussion. It would be beneficial to highlight the significance of these molecular mechanisms in the context of cancer treatment.

15.         The discussion on ANT1's role in both proapoptotic and antiapoptotic stimuli is insightful. However, to enhance clarity, consider providing a more explicit conclusion or synthesis of these seemingly contradictory roles. Discuss how this ambiguity might impact the development of therapeutic strategies targeting ANT1.
